# Nitrogen enrichment stimulates wetland plant responses whereas salt amendments alter sediment microbial communities and biogeochemical responses

**Mary Donato[1], Olivia Johnson[1], Blaire Steven[1,2], Beth A. Lawrence[1,3]***

**1** Department of Natural Resources, University of Connecticut, Storrs, Connecticut, United States of America, **2** Department of Environmental Sciences, Connecticut Agricultural Experiment Station, New Haven, Connecticut, United States of America, **3** Center for Environmental Sciences and Engineering, University of Connecticut, Storrs, Connecticut, United States of America

* beth.lawrence@uconn.edu

**Data Availability Statement:** All raw sequence datasets are available in the NCBI Short Read Archive (SRA) under the BioProject ID

## Abstract

Freshwater wetlands of the temperate north are exposed to a range of pollutants that may alter their function, including nitrogen (N)-rich agricultural and urban runoff, seawater intrusion, and road salt contamination, though it is largely unknown how these drivers of change interact with the vegetation to affect wetland carbon (C) fluxes and microbial communities. We implemented a full factorial mesocosm (378.5 L tanks) experiment investigating C-related responses to three common wetland plants of eastern North America (*Phragmites australis*, *Spartina pectinata*, *Typha latifolia*), and four water quality treatments (fresh water control, N, road salt, sea salt). During the 2017 growing season, we quantified carbon dioxide ($CO_2$) and methane ($CH_4$) fluxes, above- and below-ground biomass, root porosity, light penetration, pore water chemistry ($NH_4^+$, $NO_3^-$, $SO_4^{-2}$, $Cl^-$, DOC), soil C mineralization, as well as sediment microbial communities via 16S rRNA gene sequencing. Relative to freshwater controls, N enrichment stimulated plant biomass, which in turn increased $CO_2$ uptake and reduced light penetration, especially in *Spartina* stands. Root porosity was not affected by water quality, but was positively correlated with $CH_4$ emissions, suggesting that plants can be important conduits for $CH_4$ from anoxic sediment to the atmosphere. Sediment microbial composition was largely unaffected by N addition, whereas salt amendments induced structural shifts, reduced sediment community diversity, and reduced C mineralization rates, presumably due to osmotic stress. Methane emissions were suppressed by sea salt, but not road salt, providing evidence for the additional chemical control ($SO_4^{-2}$ availability) on this microbial-mediated process. Thus, N may have stimulated plant activity while salting treatments preferentially enriched specific microbial populations. Together our findings underpin the utility of combining plant and microbial responses, and highlight the need for more integrative studies to predict the consequences of a changing environment on freshwater wetlands.

PRJNA604015: https://www.ncbi.nlm.nih.gov/bioproject/?term=PRJNA604015. All other data will be available after acceptance in the Dryad database.

**Funding:** MD received a Summer Undergraduate Research Fellowship from the University of Connecticut (https://ugradresearch.uconn.edu/surf/) and a Micheal LeFor grant from the Connecticut Association of Wetland Scientists (http://www.ctwetlands.org/mehrhoffgrant.html). BAL received a USDA National Institute of Food and Agriculture McIntire Stennis Grant (CONS00968; https://nifa.usda.gov/program/mcintire-stennis-capacity-grant) and a US Environmental Protection Agency award (LI96172701; https://www.epa.gov/grants). BS was supported by the USDA National Institute of Food and Agriculture Hatch project 1006211 (https://nifa.usda.gov/program/hatch-act-1887). The funders had no role in study design, data collection and analysis, decision to publish, or preparation of the manuscript.

**Competing interests:** The authors have declared that no competing interests exist.

## Introduction

Wetlands play a disproportionate role in the global carbon (C) cycle; despite covering only 5–9% of the world's land surface [1], they store up to a third of terrestrial soil C [2,3] and contribute more than a third of global methane ($CH_4$) emissions, a potent greenhouse gas with 28-times the warming effect of $CO_2$ [4]. These highly productive ecosystems are increasingly dominated by monotypic graminoids [5] and have saturated soils that are key sites for anaerobic microbial processes. However, we currently have minimal understanding of how degraded water quality associated with anthropogenic activities affects the interactions among plant and microbial communities underlying wetland C processes.

Traits of dominant wetland macrophytes play an important role in wetland C cycling. Biomass production largely determines $CO_2$ assimilation rates and is often positively correlated with $CH_4$ emissions [6,7]. Plant allocation of resources belowground provides organic substrates to sediment microbial communities for anaerobic respiration [8,9], which can promote methanogenesis and increase $CH_4$ emissions [10]. However, the relationship between biomass and $CH_4$ emissions may not be so straightforward, as porous tissues of wetland plants (i.e., aerenchyma) link anoxic soil to the atmosphere; this could reduce net $CH_4$ emissions by promoting soil oxygenation via root-soil gas exchange [11,12], or increase net emissions by allowing $CH_4$ produced in underlying anoxic sediment to bypass oxidized surface sediments and waters [13,14]. Because root porosity varies among plant species and appears to be a plastic trait [15,16], we need to further elucidate its role in $CH_4$ emissions among common wetland plants subjected to impaired water quality.

Increasingly in the Anthropocene, wetland structure and function is determined by water quality because wetlands are "landscape sinks" [5] that accumulate materials and pollutants (e.g., nitrogen (N), salts) from watershed disturbances. Macrophytes such as species in the genera *Phragmites*, *Spartina*, and *Typha* are well suited to invade and dominate wetlands [5, 17,18], thus changes in water quality associated with N enrichment or salt intrusion may give these plants a competitive advantage and indirectly affect C fluxes. For example, *Phragmites australis* is a salt-tolerant invader of brackish marshes and roadsides of eastern North America that tends to create large productive monocultures that have higher $CH_4$ emissions than native communities [19,20]. Similarly, N enrichment common in agricultural and urban landscapes promotes *Typha* dominance [21], whose invasion can increase soil $CH_4$ emissions [7]. Nitrogen enrichment promotes biomass production [22] with associated increases in $CO_2$ uptake, rhizosphere oxidation, C exudation, and microbial activity [23]. The consequent effects on $CH_4$ emissions are therefore mixed; in addition to the nuanced balance of oxygen and C inputs from increased biomass, the direct effect of increased N could favor other microbes over methanogens.

Elevated salinity associated with seawater intrusion and road deicing salts can induce osmotic stress, altering growth and composition of plant and microbial communities [24,25]. Further, saline conditions change the availability of terminal electron acceptors [26], and promote organic matter flocculation [27,28], which alter microbial respiration rates. Intrusion of sulfate-rich seawater into freshwater wetlands reduces soil $CH_4$ emissions, as sulfate reduction can be thermodynamically favored over methanogenesis [29,30]. Exponential usage of deicing salts, largely sodium chloride (NaCl) throughout the temperate north [31–33], has had severe ecological consequences [34,35]. Where water residence times are high, elevated concentrations of $Na^+$ can displace other cations ($NH_4^+$, $Ca^+$, $K^+$, $Mg^+$) through cation exchange [36], causing negative effects on biotic communities due to salt stress and altered nutrient availability [37,38]. However, the consequences of road salt pollution on wetland C emissions are less well understood, and may differ from those of seawater intrusion.

Our objective was to investigate how dominant wetland plants and common water quality impairments interact to alter components of freshwater wetland C cycling. We conducted a wetland mesocosm experiment during the 2016–2017 growing seasons to test how traits (i.e., biomass, root porosity) of three common wetland plants (*Phragmites australis*, *Spartina pectinata*, *Typha latifolia*, hereafter *Phragmites*, *Spartina*, *Typha*, respectively) and four water quality treatments (freshwater control, N, road salt, sea salt) interact to alter C gas fluxes ($CO_2$, $CH_4$, C mineralization) and sediment microbial communities.

## Materials and methods

### Experimental design

We implemented an outdoor mesocosm experiment at the University of Connecticut (Storrs, Connecticut, USA), consisting of 48 mesocosms which were 378.5 L plastic tanks (79 cm x 64 cm x 132 cm; Freeland Poly-Tuf Tank©; S1A Fig). In spring 2016, we planted monocultures of three wetland plant species (*Phragmites*, *Spartina*, *Typha*), and in 2017 we implemented four water quality treatments (freshwater control, N, road salt sea salt). We replicated each plant species-water quality treatment combination four-fold and randomly assigned treatments to the 48 mesocosms. We chose common wetland plants that occur throughout eastern North America and that vary in root porosity [16], biomass production, and salt tolerance; *Phragmites* and *Typha* tend to dominate fresh to brackish marshes, whereas *Spartina* is typically considered a freshwater grass, but occurs along the upland fringes of coastal marshes in eastern North America.

We filled the bottom of each mesocosm with 15 cm of sand, and then added 30 cm of commercially screened topsoil. In June 2016, we planted four, four-month old seedlings into each mesocosm; we grew plants in a greenhouse using locally-collected, cold-stratified seed during spring 2016. Seedlings were allowed to establish during the 2016 growing season and were regularly watered to maintain saturated soils. In May 2017 we inoculated each mesocosm with 19 L of sediment collected from a nearby constructed freshwater wetland known to have methanogenic activity [39]. Water levels were maintained at 5 to 10 cm above the soil surface during the growing season (May-September) using ground water from a nearby well (pH: 7.12); water levels occasionally exceeded 10 cm after major rain events, but we ensured that water levels were consistent across tanks. Mesocosms were drained October-April when plants were dormant to prevent cracking of the plastic tubs during freezing conditions.

**Water quality treatments.** Water quality treatments (freshwater control, N, road salt, sea salt) were applied twice in 2017 (May, June). Powder forms of N and salt compounds were added to 1 L Nalgene bottles with 0.9 L of deionized (DI) water and shaken manually until fully dissolved. Once dissolved, solutions were poured evenly across assigned mesocosms; controls received 1 L of DI water. For the N treatment, we applied ammonium nitrate ($NH_4NO_3$) at a rate of 15 g N/year (two applications of 21.4 g of $NH_4NO_3$). We targeted a salinity of ~2 ppt for the two salt treatments, and during two application events added 300 g/year of dissolved salt (road salt: Diamond Crystal Winter Melt NaCl; sea salt: Instant Ocean® Sea Salt). Instant Ocean® is a commonly used saltwater aquarium additive with a similar chemical composition to seawater [40] and has been used to simulate seawater intrusion in other studies [41,42]. Treatment concentrations were selected based on previous experiments [43–45] as well as field measurements of salinity concentrations in Connecticut road-adjacent wetlands [38].

### Response metrics & analysis

**Carbon fluxes.** We measured C fluxes during three sampling campaigns in 2017 (mid-July, August, and September; approximately one, two, and three months after the last dosing treatment). We used a Picarro G2201-*i* cavity ring-down spectrometer (Picarro Inc., Santa

Clara, CA, USA) that measures $CO_2$ and $CH_4$ gas concentrations in real time (approximately every 3 s). A clear sampling chamber (base: 25 cm x 25 cm, height: 100 cm or 150 cm tall, depending on vegetation height) made of UV-resistant PVC film, and fitted with a vent tube, a sample port, and a fan to mix chamber air, was placed over a random quadrant of each meso-cosm (S1B Fig; the individual pictured in S1B Fig has provided written informed consent to publish their image alongside the manuscript). We connected the chamber to the Picarro gas analyzer via Swagelok® connections and Tygon® tubing, and deployed chambers for 10-minute incubations during daylight hours (10:00 to 16:00); an iButton temperature sensor (Maxim Integrated, San Jose, CA, USA) recorded in-chamber air temperature once every min-ute. Barometric pressure and ambient air temperature were also recorded, using a Kestrel 2500 Weather Meter (Nielsen-Kellerman, Boothwyn, PA, USA). Gas concentration measurements were corrected for the ideal gas law using temperature, pressure, and chamber volume. Flux rates were calculated based on linear changes in gas concentrations over time if $R^2$ values were $> 0.85$. For rates with $R^2$ values $< 0.85$, we visually inspected plots of concentration vs. time; rates that exhibited evidence or record of equipment malfunction (chamber tipping, etc.) or ebullition were removed from analysis (n = 5). If the linear regression of time vs. gas con-centration did not differ from zero, we assigned the gas flux as zero.

**Plant biomass & root porosity.** We estimated aboveground biomass using species-specific allometric equations developed from ~50 oven-dried (65°C) stems of each species in 2016, relat-ing stem height to dry biomass; all equations were second order polynomials (*Typha*: $R^2 = 0.92$, 95% CI = ± 0.01 g; *Spartina*: $R^2 = 0.92$, 95% CI = ± 0.006 g; *Phragmites*: $R^2 = 0.94$, 95% CI = ± 0.004 g). All stem heights were measured in September 2017 to estimate aboveground biomass for each species-water quality treatment. Photosynthetically active radiation (PAR) measure-ments were taken at this time using a Decagon LP-80 Ceptometer (Decagon Devices, Pullman, WA, USA). Three measurements per mesocosm were averaged above the plant canopy and at the sediment surface to estimate the fraction of PAR (fPAR) transmitted through the canopy.

We installed in-growth root cores to measure 2017 root production in each tank (May- Sep-tember 2017) [46]. Nylon mesh cylinders (5-cm diameter x 13-cm long) with a plastic base were packed with screened, root-free topsoil (same as that used to fill mesocosms) to a similar bulk density as the surrounding soil (~1.8 g dry soil/$cm^3$, average of 2016 soils). In-growth cores were installed into excavated holes of similar dimensions in May and were removed from the tanks by cutting around the outside of the core with a serrated knife and pulling it free of the tank sediment in September. Each core was emptied into a 2-mm sieve and soil was washed away with a garden hose to isolate the roots.

In the lab, we identified three root segments (~5 cm in length) per core that were elastic and light or white in color to estimate root porosity using methods similar to [15]. We blotted excess water from the outside of the roots with lab tissues then individually weighed each seg-ment on a microbalance. To keep roots submerged under 500 mL of water in a 1 L side arm flask, we attached a paper clip to each root segment. The flask was attached to a vacuum pump for five minutes to replace all of the airspace in the root with water. The roots were removed from the water and weighed again. The difference of the two weights divided by the initial weight estimates the proportion of the root mass that was originally airspace. Roots sampled for porosity were then returned to the bulk root sample from each mesocosm, dried ≧ 72 hours at 60°C, and weighed. Root porosity estimates for each mesocosm were averaged and then multiplied by belowground biomass to calculate total porosity. Belowground biomass estimates were calculated by scaling the mass of roots in the area of the ingrowth core to 1 $m^2$. Likewise, aboveground biomass estimates were scaled to units of g/$m^2$.

**Water chemistry.** We constructed wells to monitor and sample pore water chemistry. We cut 30-cm sections of PVC pipe (2.54-cm diameter), capped the bottom, sliced narrow slits to

7 cm, and wrapped 1-mm nylon screen around the slitted area to limit sediment intrusion. We pounded wells into the center of each tank to 15-cm depth. Conductivity, salinity, and pore-water temperature measurements were taken during each gas sampling event using a YSI Eco-Sense® EC300A meter (YSI Incorporated, Yellow Springs, OH, USA). Pore water samples were taken from each mesocosm for analysis at the end of the growing season in September 2017. Pore water wells were purged and then water samples were collected with a nylon syringe and tubing and placed into acid-washed 50 mL centrifuge tubes. Samples were stored at 4˚C until analysis. Water samples were centrifuged and filtered using 110-mm Whatman G/FF paper filters and analyzed for nitrate ($NO_3^-$) and ammonia ($NH_4^+$) on a SmartChem®200 discrete analyzer (Westco Scientific Instruments, Brookfield, CT, USA). Whatman G/FF-filtered samples were quantified for total organic carbon (TOC) on a Shimadzu Total Organic Carbon Analyzer using EPA Method 415.1, and sulfate ($SO_4^{-2}$) and chloride ($Cl^-$) on a Dionex Ion Chromatography System-1100 (Thermo Fisher Scientific, Waltham, MA, USA).

**Carbon mineralization.** Surface soil samples (5-cm diameter to 10-cm depth) were collected in September 2017 to estimate heterotrophic respiration rates. Soils were sieved through 2-mm brass screens; a 10-g subsample of sieved soil was dried at 105˚C for 48 hours to calculate soil moisture content, and a 50-g subsample was placed in a 0.95-L canning jar. Jars were attached via a 15-port manifold sampling system to the Picarro G2201-*i* and their headspace $CO_2$ concentrations were sampled for six minutes approximately every four hours over a 24-hour period [47]; $CO_2$-free air soda lime blanks were used to flush the lines between samples. We converted gas accumulation rates measured as ppm/s to umol/s using the ideal gas law. Correcting for soil moisture content, we calculated C mineralization rates as the accumulation of gas over time per gram of dry soil.

**Statistical analysis.** All statistical analyses were conducted using R Studio 1.1.419 using R 3.5.1. Data were log-transformed to improve normality of residuals and homogeneity of variances when necessary. We tested for fixed effects of plant species, water quality treatments, and their interaction on gas fluxes, biomass, total porosity, and water chemistry data using Linear Mixed Effects models (*lme* and *aov* commands), with mesocosm tanks set as the random effect. Initial analysis (*lme* command) indicated consistent treatment responses across our three sampling campaigns for both $CO_2$ ($F_{2,94} = 0.83$, $p = 0.439$) and $CH_4$ fluxes ($F_{2,87} = 2.64$, $p = 0.077$), thus we aggregated carbon flux data from the three sampling campaigns to test for differences among vegetation and water quality treatments. We used the September gas sampling campaign data to investigate correlations with other variables as it aligned temporally with when we collected biomass and pore water chemistry data. Correlations between explanatory and response variables were analyzed using the *cor.test* command; we used Pearson's correlation coefficient (r) for parametric data and Spearman rank correlation coefficients ($r_s$) for non-parametric data (i.e., when transformations did not improve normality). Means ± 1 SE are reported.

**Sediment microbial characterization.** Approximately 5 g soil samples were collected from the upper 10 cm of sediments using an ethanol-sterilized spoon. Samples were collected in the vicinity of the plants which contained a large amount of root material, however these were bulk soil samples from the root zone, not specifically rhizosphere soils. Soil samples were placed in sterile Whirl-pak bags, flash frozen on dry ice, and stored at -80˚C until further processing.

DNA was extracted from ~1 g of sediment using the DNeasy PowerSoil kit (Qiagen) using the manufacturer's protocols with the exception that bead mill beating was performed on a Retch MM301 Ball Mill (30 hz for 1 minute). The V4 region of bacterial 16S rRNA genes was amplified using primers 515F and 806R with Illumina adapters and dual indices (8 basepair golay on 3' [48], and 8 basepair on the 5' [49]). The amplification products were sequenced at UConn's MARS (Microbial Analysis, Resources, and Services) Illumina MiSeq platform.

Demultiplexed sequences were assembled into contigs and quality screened in the mothur software package (version 1.41.1.5; [50]). All sequences were selected to be at least 255 bp in length, contain no ambiguous bases, and no homopolymers of more than 8 bp. Chimeric sequences were identified with the mothur implementation of VSEARCH [51], and all potentially chimeric sequences were removed. Sequences were clustered into operational taxonomic units (OTUs) using a 100% sequence identity threshold, employing the OptiClust algorithm in mothur [52]. Taxonomic classification of sequences was performed with the Naïve Bayesian classifier [53] against the SILVA reference alignment (release 132) [54] in the mothur software package.

Prior to determining alpha-diversity via the nonparametric Shannon's diversity index (H'), data-sets were randomly subsampled to the size of the smallest dataset (omitting outliers with <1000 sequences), resulting in 5,720 sequences per dataset. Significant differences in OTU relative abundance were tested for with the ALDEx2 package. Prior to identifying significant differences, OTU count data were transformed using the centered log-ratio and normalized through Monte Carlo sampling with Bayesian sampling of 128 Dirichlet instances [55]. Both the Kruskal-Wallis and generalized linear model tests were performed and an OTU was considered to be significantly different in relative abundance if the p-value was ≤0.05 after adjusting for multiple testing with the Benjamini-Hochberg correction. The ternary plot of OTU relative abundance was generated with the ggtern extension package in R [56]. All raw sequence datasets are available in the NCBI Short Read Archive (SRA) under the BioProject ID PRJNA604015.

## Results

### Plant biomass and root porosity responses

Biomass production differed among vegetation above- and belowground (above: $F_{2,36} = 46.5$, $p < 0.001$; below: $F_{2,36} = 6.8$, $p = 0.003$), as well as among water quality treatments aboveground (above: $F_{3,36} = 144.0$, $p < 0.001$). However, we observed interactions between species and water quality treatment for aboveground biomass ($F_{6,36} = 6.5$, $p < 0.001$), principally because *Spartina* aboveground biomass responded strongly to N enrichment (Fig 1). fPAR transmission was strongly negatively correlated with aboveground biomass ($r = -0.80$, $p < 0.001$), and differed among species ($F_{2,42} = 3.8$, $p = 0.030$) and water quality treatments ($F_{3,42} = 24.3$, $p < 0.001$), with greater transmission through *Typha* (71.3% ± 1.9) than *Spartina* canopies (64.6% ± 3.5). Nitrogen enrichment reduced fPAR (N: 52.9% ± 2.9) relative to the control and salt treatments (72.5% ± 1.1).

Root porosity did not differ across water quality treatments ($F_{2,36} = 1.85$, $p = 0.15$), but *Spartina* roots were more porous ($F_{2,36} = 7.6$, $p < 0.05$; Table 2) and had greater total root porosity ($F_{2,36} = 6.08$, $p < 0.05$; Table 1) than the other two species (Table 1).

### Carbon fluxes

We observed differences in $CO_2$ uptake among species ($F_{2,36} = 34.27$ $p < 0.0001$) and water quality treatments ($F_{3,36} = 12.48$, $p < 0.0001$), but did not observe an interactive effect of species and water quality treatment ($F_{6,36} = 1.12$, $p = 0.369$). *Spartina* (36,239 ± 2744 µmol m$^{-2}$ hr$^{-1}$) and *Typha* (23,880 ± 1503 µmol m$^{-2}$ hr$^{-1}$) had greater $CO_2$ uptake than *Phragmites* (14,070 ± 1112 µmol m$^{-2}$ hr$^{-1}$; Fig 2A). Nitrogen addition (39,564 ± 3467 µmol m$^{-2}$ hr$^{-1}$) increased $CO_2$ uptake relative to freshwater controls (19,392 ± 1618 µmol m$^{-2}$ hr$^{-1}$) and sea salt treatments (19,751 ± 1990 µmol m$^{-2}$ hr$^{-1}$), but did not lead to different $CO_2$ uptake from our road salt treatment (20,211 ± 1467 µmol m$^{-2}$ hr$^{-1}$; Fig 2B).

Methane emissions differed strongly among vegetation species ($F_{3,36} = 40.83$, $p < 0.0001$; Fig 3A); *Spartina* (101.9 ± 12.4 µmol m$^{-2}$ hr$^{-1}$) had the highest $CH_4$ fluxes, followed by *Typha*

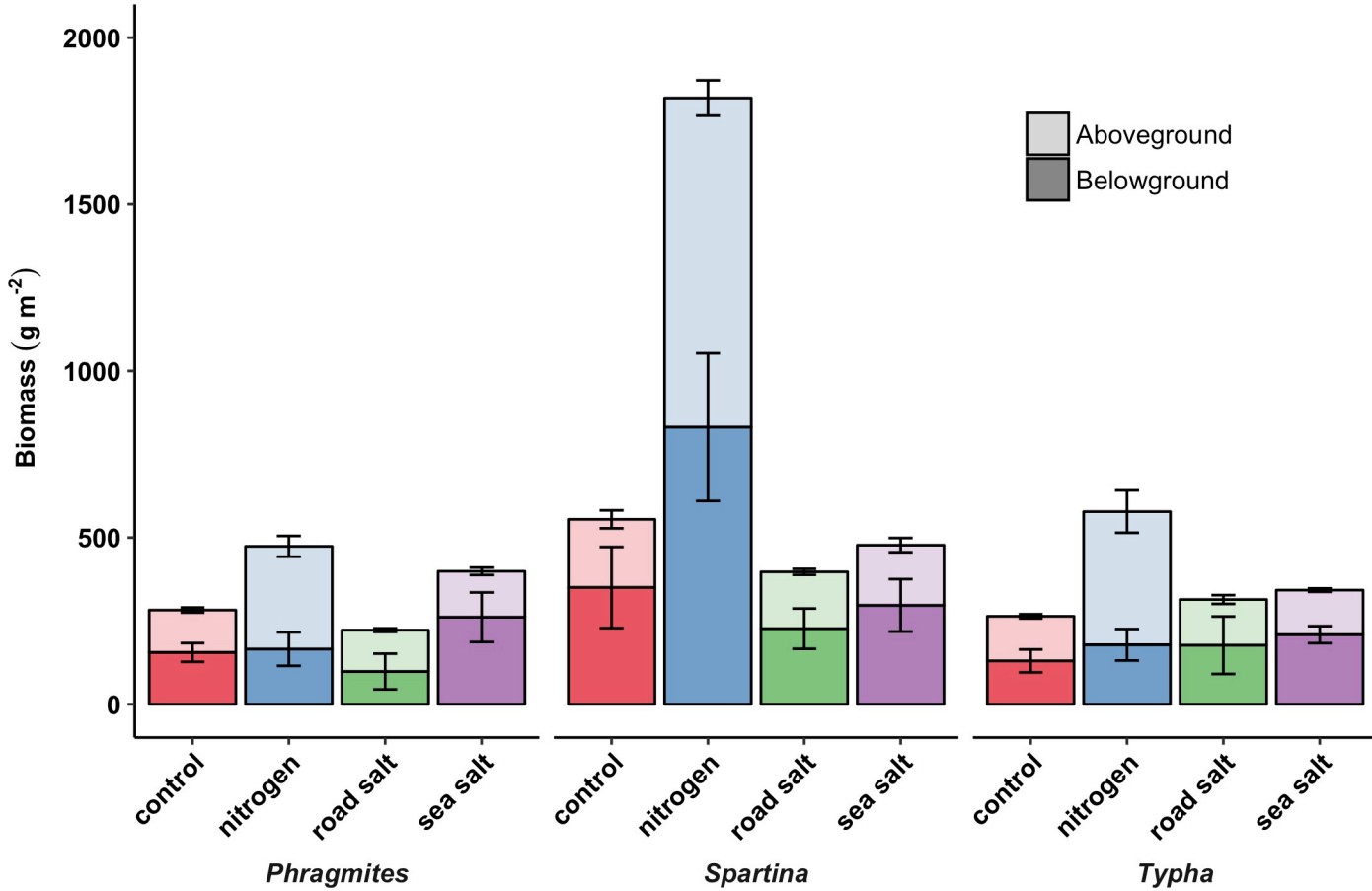

**Fig 1. Biomass allocation by vegetation species-water quality treatment combinations.** Mean (± SE) above- and belowground biomass by species-water quality combinations in 2017 in a full factorial mesocosm experiment where each treatment combination was replicated four-fold.

(61.8 ± 10.7 μmol m$^{-2}$ hr$^{-1}$), and then *Phragmites* (21.0 ± 2.7 μmol m$^{-2}$ hr$^{-1}$). We also observed differences among water quality treatments ($F_{3,36}$ = 6.31, p = 0.002; Fig 3B), with sea salt addition halving CH$_4$ emissions (34.7 ± 5.8 μmol m$^{-2}$ hr$^{-1}$) relative to the other water quality treatments (70.3 ± 7.7 μmol m$^{-2}$ hr$^{-1}$). Water quality treatment effects were consistent across vegetation species, as we did not observe an interaction among these factors ($F_{6,36}$ = 1.10, p = 0.383).

## Pore water chemistry

Pore water chemistry was generally more responsive to water quality than vegetation treatments (Table 2), though SO$_4^{-2}$ and DOC differed among species, with *Spartina* having lower

**Table 1. Root porosity differed among vegetation species.**

| Vegetation species | Root porosity (%) | Total root porosity |
|---|---|---|
| *Phragmites* | 25.5[a] ± 2.5 | 45.9[a] ± 8.8 |
| *Spartina* | 35.1[b] ± 3.1 | 137.1[b] ± 24.8 |
| *Typha* | 25.3[a] ± 2.5 | 46.6[a] ± 9.4 |

Average (± 1 SE) 2017 root porosity (measured from 3 root subsamples per mesocosm) and total root porosity (% porosity x total belowground biomass) for each plant species (n = 16). Superscripts indicate significant differences between vegetation species after TukeyHSD post-hoc comparisons.

**Table 2. Pore water chemistry ANOVA results.**

| | Vegetation | | | Water Quality | | |
|---|---|---|---|---|---|---|
| **Response** | **df** | **F** | **p** | **df** | **F** | **p** |
| $SO_4^{-2}$ | 2, 41 | 6.3 | **0.004** | 3, 41 | 43.3 | **<0.001** |
| $Cl^-$ | 2, 40 | 0.2 | 0.847 | 3, 40 | 94.7 | **<0.001** |
| $NO_3^-$ | 2, 13 | 1.9 | 0.194 | 3, 13 | 0.9 | 0.434 |
| $NH_4^+$ | 2, 36 | 0.1 | 0.903 | 3, 36 | 1.4 | 0.236 |
| DOC | 2, 38 | 6.2 | **0.005** | 3, 38 | 70.9 | **<0.001** |

Two-way ANOVA results that tested how pore water chemistry differed among vegetation and water quality treatments. Note that 29 $NO_3^-$ samples were below instrument detection limit, resulting in low sample size.

concentrations ($SO_4^{-2}$: 1.03 ± 0.21 mg/L; DOC: 8.00 ± 1.25 mg/L) than *Typha* and *Phragmites* ($SO_4^{-2}$: 2.28 ± 0.53 mg/L; DOC: 10.14 ± 1.43 mg/L). Salt ions associated with experimental treatments differed as expected: $Cl^-$ concentrations were much greater with road and sea salt addition (89.2 ± 4.6 mg/L) than control and N-enriched treatments (1.8 ± 0.5 mg/L), and sea salt treatment doubled $SO_4^{-2}$ concentrations (3.2 ± 0.5 mg/L) compared to the other treatments (1.4 ± 0.9 mg/L). We did not observe treatment differences in $NO_3^-$ nor $NH_4^+$ concentrations (Table 2); $NO_3^-$ concentrations averaged 0.22 ± 0.05 mg/L among the 19 samples that were above our instrument's detection limit, whereas $NH_4^+$ concentrations averaged 1.00 ± 0.53 mg/L. Salt addition reduced pore water DOC concentrations, as we observed three times

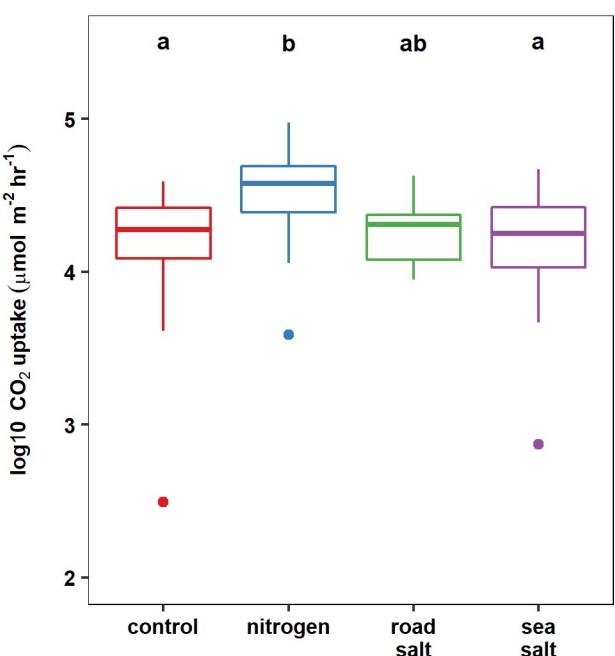

**Fig 2. $CO_2$ uptake differed among vegetation and water quality treatments.** Boxplots of 2017 log-transformed $CO_2$ uptake rates (samples pooled across July, August, September sampling campaigns) by (a) vegetation and (b) water quality treatments. Note that measurements were estimates of net ecosystem exchange, integrating photosynthetic uptake, auto and heterotrophic respiration from transparent chambers. Differences between groups are indicated by non-overlap of letters, based on post-hoc Tukey contrasts.

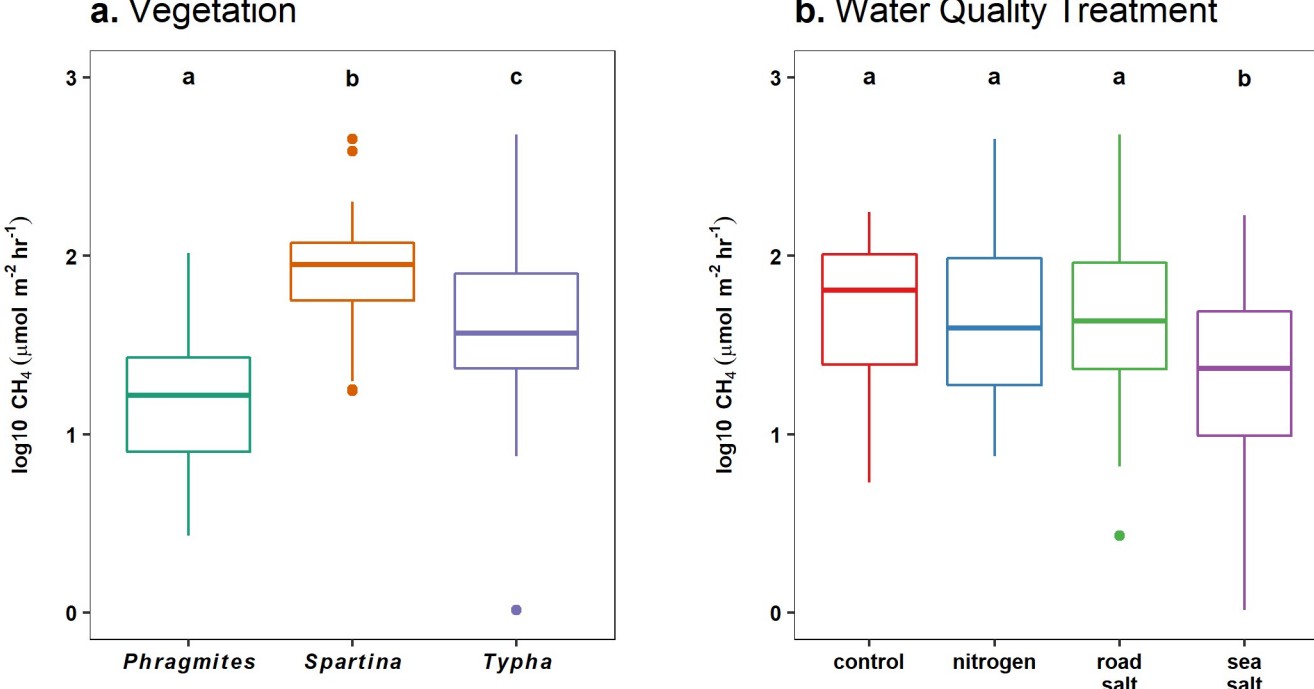

**Fig 3. $CH_4$ emissions by vegetation and water quality treatments.** 2017 $CH_4$ emissions from July, August, and September sampling campaigns were pooled and log transformed to examine differences among (a) vegetation and (b) water quality treatments. Differences between groups are indicated by non-overlap of letters, based on post-hoc Tukey contrasts.

as much DOC in control and N-enriched mesocosms (14.9 ± 1.0 mg/L) than in road and sea salt tanks (5.0 ± 0.4 mg/L).

## Carbon mineralization

Soil C mineralization rates did not differ among vegetation ($F_{2,38}$ = 2.4, p = 0.11) but were reduced with sea and road salt compared to freshwater controls and N enrichment ($F_{3,38}$ = 11.2, p < 0.001) (Fig 4).

## Correlations with carbon fluxes

Aboveground biomass was positively correlated to $CO_2$ uptake (r = 0.60, p < 0.0001), but $CH_4$ emissions were not correlated with aboveground, belowground, nor total biomass. However, total root porosity was positively correlated with $CH_4$ emissions (r = 0.38, p = 0.008). Pore water chemistry associated with our salt treatments ($SO_4^{-2}$, $Cl^-$) influenced several C responses. We observed negative correlations between $SO_4^{-2}$ concentration and $CH_4$ emissions ($r_s$ = -0.337, p = 0.024), between $Cl^-$ concentrations and C mineralization ($r_s$ = -0.577, p = < 0.0001), and between DOC and C mineralization ($r_s$ = -0.769, p = < 0.0001).

## Bacterial community composition

Cluster Canonical Correlation Analysis (cluster-CCA) was used to investigate the relationship between bacterial 16S rRNA gene sequence datasets (Fig 5). Clustering by vegetation showed substantial overlap in community composition, although there was a significant difference in centroids between vegetation types (Fig 5A; p = 0.005). Clustering was more apparent when aggregated by water quality treatment, with the salt treatments separating distinctly from the

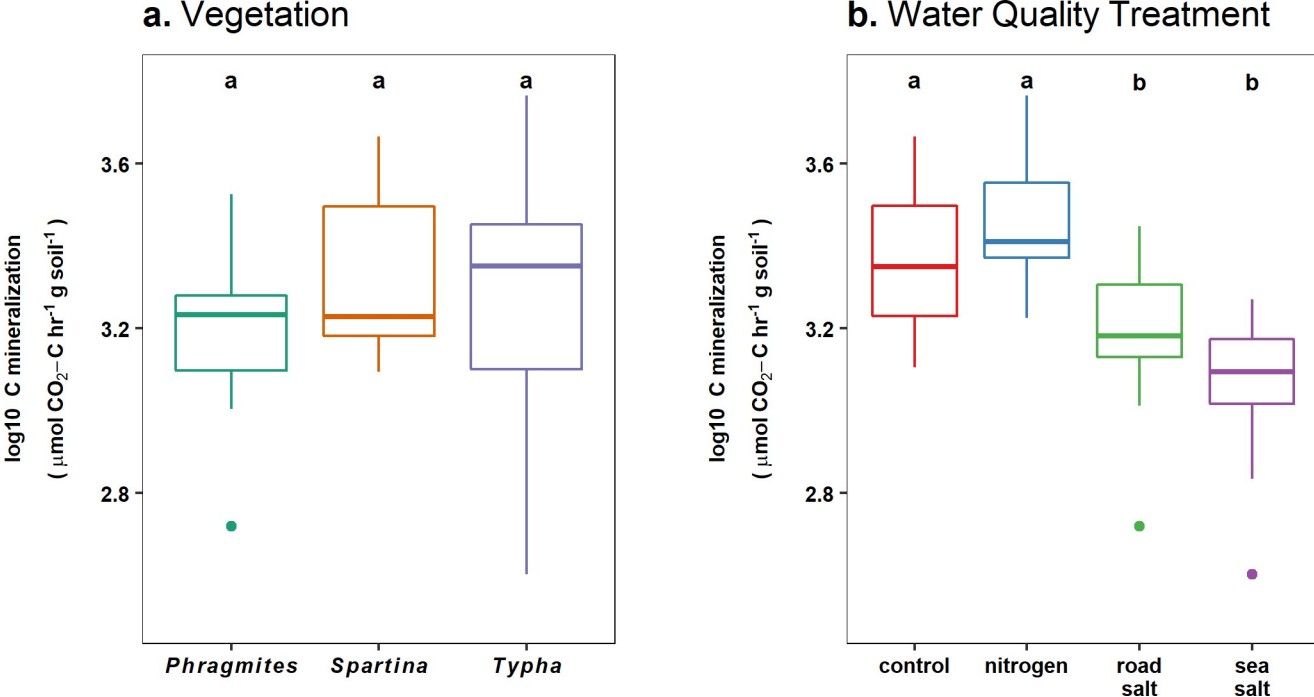

**Fig 4. Carbon mineralization rates by vegetation and water quality treatments.** Log-transformed sediment C mineralization rates estimated using 24-hour laboratory incubations did not differ among (a) vegetation, but differed among (b) water quality treatments. Differences between groups are indicated by non-overlap of letters, based on post-hoc Tukey contrasts.

control and N amendments (Fig 5B). However, there was no significant clustering that differentiated the controls and the N enrichment or between the two salting treatments (road or sea salt).

**Bacterial diversity.** Bacterial diversity was assessed by calculating the non-parametric Shannon's diversity index. When the datasets were clustered by vegetation, *Typha* showed the highest average diversity, with the lowest diversity amongst *Phragmites* (Fig 6A). Water quality treatment showed a clear decrease in diversity associated with the salt treatments (Fig 6B).

**Differentially abundant OTUs due to vegetation.** The abundance of the numerically dominant OTUs were plotted as a ternary diagram to display their relative abundance among the three plant species (Fig 7A). Most OTUs belonged to five bacterial phyla, with the Proteobacteria being most common. The majority of the OTUs were present at similar relative abundances among the three vegetation types as evidenced by their clustering in the center of the ternary diagram (Fig 7A). Only two OTUs were identified as significantly different in relative abundance, and their abundances in each vegetation type is displayed in Fig 7B. Otu000028 was classified to the genus *Geobacter* (Phylum, Proteobacteria) and was enriched in the *Spartina* mesocosms. In contrast Otu000322, classified to the *Novosphingobium* (Phylum, Proteobacteria), was uniquely present in with *Typha*. Generally, these data indicate that OTU relative abundance was sensitive to the different plant species, with only a very limited number of OTUs showing a shift in relative abundance in response to plant species.

**Differentially abundant OTUs due to water quality treatment.** OTU relative abundance in the controls was plotted against the treatments to test for shifts in abundance due to the different amendments. OTUs were present in similar relative abundance between control and N enrichment treatments, and no OTUs were identified as significantly different in relative abundance due to N (Fig 8A). In comparison, multiple OTUs were identified as differentially

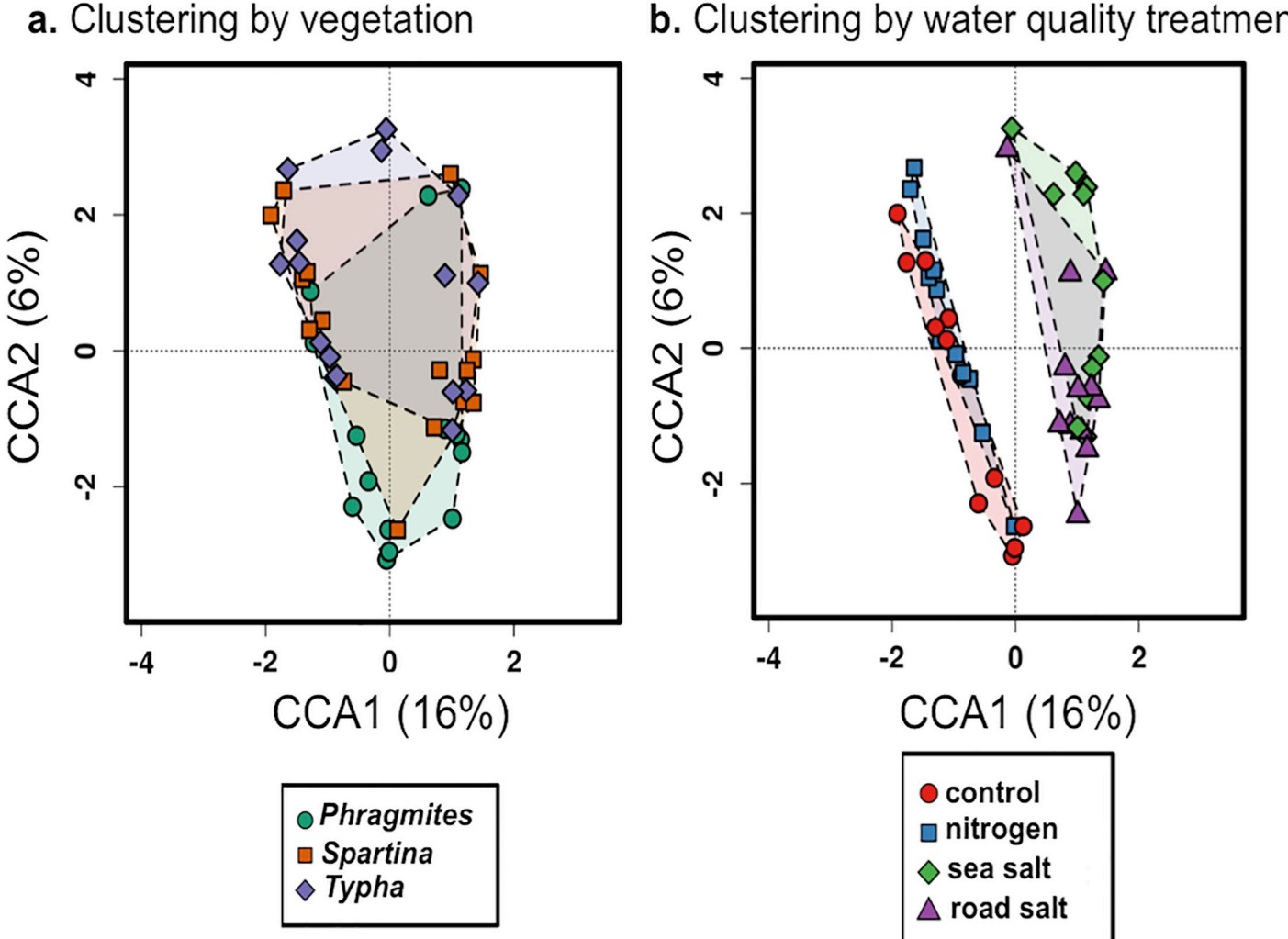

**Fig 5. CCA-clustering of bacterial 16S rRNA gene datasets.** (a) Data clustered by vegetation type. Significance of clustering was tested with the permutation anova CCA test and vegetation was a significant factor for clustering (p = 0.005). (b) The same data clustered by water quality treatment, which was also a significant factor in dataset clustering (p = 0.001).

abundant due to the road and sea salt treatments. We further determined if the differentially abundant OTUs from the two salt treatments were unique or common to each condition (Fig 8B). A total of 86 OTUs were identified as significantly different of which 25 (29%) were common to both the road salt and sea salt treatments. In this regard, there appears to be a set of OTUs that share a similar response to salt, irrespective of the source. The differentially abundant OTUs identified as common to both salt treatments were predominantly within the phylum Proteobacteria (Fig 8C).

Finally, we investigated those OTUs that were enriched in controls versus those that were enriched with salt and were shared between both the road salt and sea salt treatments (S1 Table). A diverse set of OTUs were identified, belonging to six different phyla and 17 families. All of the taxa were heterotrophic groups with a variety of different growth types and strategies. For instance, an OTU related to the genus *Sideroxydans*, an iron oxidizing group of bacteria [57], was enriched in the control samples (S1 Table). In contrast, three OTUs related to the genus *Geobacter* were enriched in the salted sediments (shared in both road salt and sea salt).

## a. Vegetation

## b. Water quality treatment

**Fig 6. Bacterial diversity in sequence datasets.** (a) Diversity in datasets grouped by vegetation type. (b) Diversity in datasets grouped by water quality treatment. Significant differences between groups are indicated by non-overlap of letters, based on post-hoc Tukey contrasts.

Members of the *Geobacter* genus are thought to be the primary drivers of oxidizing organic matter coupled to the reduction of iron and manganese [58]. In this respect, these data point to a state change in the iron cycle in the mesocosms under the salt treatments, which points to a decreased availability of dissolved iron under elevated salt. The remaining OTUs largely belonged to general heterotrophic bacteria or were not able to be classified to taxonomic ranks

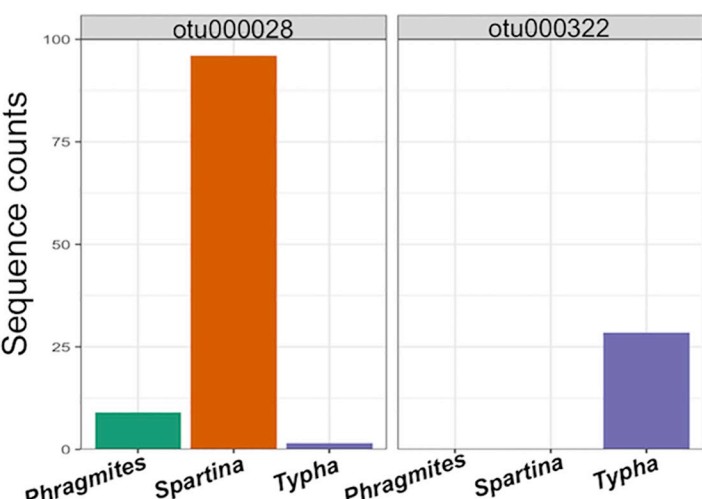

## a. Abundance of OTUs

## b. Differentially abundant OTUs

**Fig 7. OTU relative abundance in association with vegetation.** Only the 1500 most abundant OTUs are displayed. (a) Ternary diagram displaying OTU abundance among the three plant species. The two OTUs identified as significantly different in relative abundance are indicated by the arrows. (b) Median counts per sample of each of the differentially abundant OTUs. A table showing the classification of the differentially abundant OTUs is shown in S1 Table.

**Fig 8. Differentially abundant OTUs due to water quality treatment.** (a) Each point represents a detected OTU and its counts in controls versus treatment. OTUs colored in red were identified as significantly different in abundance. (b) Differentially abundant OTUs unique and shared among the two salt treatments, (c) Taxonomic classification of differentially abundant OTUs in salt treatments.

deeper than family, which limits the confidence that functional predictions can be made from these classifications.

## Discussion

Wetlands play a major role in global C dynamics, but understanding how wetland plants, sediment microbial communities, and water quality interact is currently not well resolved. To help bridge this gap, we conducted a mesocosm experiment in which we manipulated plant species (globally dominant wetland genera- *Phragmites*, *Typha*, *Spartina*) and common water quality impairments (N-enrichment, salinization via road or sea salt) to investigate C and microbial responses. We found that plant species had strong effects on our response metrics, with largely similar patterns in response to water quality treatments across plant species. However, water quality treatments appeared to have distinct effects on plant vs. microbial responses; N enrichment increased biomass production and $CO_2$ uptake, whereas salinization reduced $CH_4$

emissions (with sea salt), reduced heterotrophic respiration, altered microbial composition, and decreased microbial diversity.

## Biomass and C process responses

Rates and allocation of biomass production are the foundation of C cycling in wetlands. Not surprisingly, we observed that greater aboveground biomass promoted greater $CO_2$ uptake, and that N-enrichment amplified biomass production, particularly in *Spartina*, which had five times greater aboveground biomass production with N addition than controls. Anecdotally, we observed higher algae abundance in surface waters of *Typha* and *Phalaris* with N addition; higher levels of PAR penetrating through sparser canopies may have stimulated algal production and resulted in similar increases in $CO_2$ uptake across all vegetation treatments with N addition. Interestingly, salt addition (300 g/m$^2$/y) did not reduce biomass production compared to freshwater controls at the relatively low, but environmentally relevant, salinity levels (2 ppt) we targeted. Dramatic biomass reductions for freshwater macrophytes were observed when salinity treatments exceeded 4 ppt in [45]. Likewise, [38] observed a wetland seed bank threshold of 2 ppt for species richness, diversity, and aboveground biomass, with reductions in plant responses in NaCl treatments > 2 ppt, suggesting that common freshwater wetland plants may be resilient to salinity levels $\leq$ 2 ppt.

However, biogeochemical processes appear to be more sensitive to salinization. We observed reduced $CH_4$ emissions with $SO_4^{-2}$ rich sea salt addition; while we did not quantify how water quality treatments effected pH, under circumneutral pH, $SO_4^{-2}$ is thermodynamically favored over the reduction of C compounds [59,60]. Likewise, we observed a negative correlation between $SO_4^{-2}$ concentrations and $CH_4$ emissions across all treatments. Both salinity treatments decreased DOC concentrations, likely due to salt-induced flocculation which promotes particle aggregation [27,28], hence exclusion during filtration. We found no correlation between DOC and $CH_4$ emissions as found in other studies [61]. However, we observed decreased C mineralization rates (i.e., heterotrophic respiration) and decreased diversity of microbial communities in our salt treatments, potentially pointing to osmotic stress of certain microbial populations. Similar to [62], we did not observe an effect of N addition on C mineralization rates, indicating excess nutrients were assimilated by plants, algae or microbes in the water column, but not by soil microbes in soil.

In contrast to other studies [6,7], we did not observe positive correlations between biomass and $CH_4$ emissions, though total root porosity (% root porosity x root biomass) was positively correlated to $CH_4$ emissions. Still, our data suggest that porous plant tissue acted similarly to a straw, allowing methane produced in anoxic sediment to bypass surface oxic layers and travel into the atmosphere, as observed by others [13,63]. *Spartina* had greater total root porosity than the other two species, providing a large pathway for $CH_4$ to escape to the atmosphere. *Spartina* also had lower porewater $SO_4^{-2}$ concentrations than other species; thus, elevated $CH_4$ emissions from *Spartina* would be expected, as these conditions may favor methanogenesis [59]. Why *Spartina* had lower porewater $SO_4^{-2}$ concentrations is unclear, however, as rhizospheric oxygenation should decrease sulfate reduction, thereby maintaining large $SO_4^{-2}$ pools. Elevated uptake of $SO_4^{-2}$ by *Spartina* is plausible, as [64] observed differential species uptake.

## Microbial community response

Plant species played a significant, if small role in sediment microbial community composition in our study (Fig 5). The rhizosphere of wetland plants, the zone of soil directly in contact with the plant root harbors elevated bacterial activity and altered communities in comparison to bulk soils [65,66]. While we did not specifically isolate rhizospheric soils, we observed

differences in Shannon's diversity among vegetation treatments in bulk soils, suggesting the pervasive influence of vegetation on soil bacteria. However, the majority of the identified bacterial OTUs were present in all three vegetation treatments in similar proportions, with only two OTU's identified as significantly different in relative abundance among the three plant species (Fig 7). Thus, most bacteria appeared largely indifferent to plant species. In this regard, the influence of the plant on sediment microbial communities may be mostly limited to sediments in direct contact with roots. Instead, sediment microbial communities appeared to respond to changes in sediment properties, particularly those associated with salting, such as osmotic stress.

The salt treatments induced a reduction in the diversity of the sediment microbial communities (Fig 6B). Furthermore, a substantial fraction of the bacterial OTUs that shifted in relative abundance were common to both salt treatments, road salt and sea salt (Fig 8B). This suggests that the elevated osmotic stress likely affected a similar group of bacteria. However, the shifts in relative abundance due to the salt treatments were generally among the numerically rare populations, whereas the most abundant OTUs were resilient to the treatments (Fig 8C). This suggests that the dominant bacteria in the mesocosms were largely unaffected by the salt treatment. The differentially abundant OTUs did point to an alteration in the iron cycle in the sediments under the salt treatment. The enrichment of *Sideroxydans* in the control mesocosms in comparison to an enrichment of *Geobacter* with elevated salt suggests a shift from iron oxidation to iron reduction with the addition of salt. Further, similar to previous experimental findings [62], we observed reduced mineralization of labile carbon from the salt treatments, which may have been associated with reduced microbial diversity or shifts in community composition. Thus, the osmotic or redox stress induced by the salt treatment did appear to shift biogeochemical cycles in the sediments.

We hypothesized that we would observe a unique set of bacteria enriched in the sea salt treatment. This is because the sulfates in sea-water are thought to support sulfate-reducing communities which then outcompete methanogens. We observed a reduction of methane emissions in the sea salt treatment, yet we did not observe an enrichment of sulfate reducers (S1 Table) which could indicate higher sensitivity to salinity than to redox conditions. Furthermore, none of the differentially abundant OTUs in sea salt treatments were associated with methanotrophic populations (S1 Table), bacteria capable of oxidizing methane [67]. As the primers employed in this study were designed to amplify bacterial 16S rRNA genes, they were not able to detect methanogenic archaea so we cannot directly address the effects of sea salt on methane producing populations. Thus, the sediment microbial data was not particularly predictive in the reduction of $CH_4$ observed under the sea salt treatment. However, our data only describe the composition of the sediment communities. It is possible that water quality treatments shifted the activity of particular microbial populations, such as sulfate-reducers, methanotrophs, or methanogens, without a concurrent alteration in their relative abundance. Future studies incorporating metrics of microbial activity may better address changes in the functions of the microbial community under differing water quality treatments.

### Experimental design constraints

In the field, wetland vegetation, soils, and hydrology are often confounded, so a controlled mesocosm experiment allowed us to systematically test how vegetation and water quality treatments alter a range of biological and biogeochemical responses. However, relics of our experimental design should be considered when interpreting or comparing our results with other investigations. While invasive *Phragmites* is commonly known as an extremely productive and dominant species [68–70], the *Phragmites* we used in our study was a relatively short and sparse strain that sequestered less $CO_2$ and emitted less $CH_4$ than either *Spartina* or *Typha*.

This is likely associated with the seed source we used; we collected seed from a population growing out of a groundwater seep at the base of a hill on UConn's campus, which may not be wholly representative of the species. We manipulated the hydroperiod of our mesocosms to promote reduced soils during the growing season; while draining the tanks during winter to prevent tanks from cracking may have altered microbial composition and redox conditions, consistent manipulation of soils and hydrology allows us to draw inferences about responses to our vegetation and water quality treatments. Further, though we did not measure soil pH or the oxidative state of our mesocosms, it is possible that our treatments altered these physio-chemical parameters [37,62] and indirectly effected the biogeochemical changes we observed.

## Conclusions

Wetlands are crucial landscape sinks, often occurring in low-lying areas that collect polluted or impaired runoff from surrounding watersheds, and are on the front lines of sea level rise, making them vulnerable to salt water intrusion. In turn, water quality can affect plant species composition and production rates, which are underlying drivers of wetland C cycling. Our results indicate that plant traits (biomass, root porosity) as well as species identity are important determinants of C gas flux. Particularly in areas vulnerable to invasive species and community shifts, presence or exclusion of key species has the potential to alter $CO_2$ uptake or $CH_4$ emission rates occurring within wetlands. Another important driver of C flux in freshwater wetlands is water quality. Different water quality impairments such as N, road salt, and sea salt affect C gas flux in different ways. Nitrogen enrichment's influence on biomass production and increased gas flux make it a prominent driver of change in wetlands exposed to agricultural runoff as well as wastewater. The reduction of $CH_4$ emissions due to salt-water intrusion of sea level rise exhibits the power of small water quality changes within the system. Although the relatively low concentrations of salt used in this experiment (2 ppt) did not significantly affect plant traits such as biomass production, they did alter the water and sediment chemistry enough to influence the sediment microbial communities therefore altering $CH_4$ emissions.

 The vegetation and water quality impairments used in this experiment are common throughout not only eastern North America, but also many locations worldwide. With the crucial role that wetlands play in the global C cycle, it is important to better understand the integration between plant performance and microbiology and how these factors influence C gas fluxes. We recommend further examination of the interactions among emerging contaminants that can exacerbate water quality issues [71], vegetation, and wetland C cycling.

## Supporting information

**S1 Fig. Wetland mesocosm experimental setup.** (a) A mesocosm tank experiment was set up at the University of Connecticut in 2016–2017 to test how plant species and water quality treatments influenced carbon gas fluxes and sediment microbial communities. (b) Co-author O. Johnson monitors real time C fluxes using a transparent floating chamber connected to a Picarro g2201-*i* during the 2017 growing season.
(PDF)

**S1 Table. OTUs depleted or enriched in association with the salt treatments.**
(PDF)

## Acknowledgments

Invaluable field and lab assistance was provided by Alaina Bisson, Aidan Barry, Becky Fahey, Samantha Walker, Cooper Hernsdorf, Yi Liu and Regan Huntley. We appreciate R code

shared by Elizabeth Brannon and Serena Moseman-Valtierra that we used to calculate gas flux rates.

## Author Contributions

**Conceptualization:** Mary Donato, Beth A. Lawrence.

**Data curation:** Beth A. Lawrence.

**Formal analysis:** Beth A. Lawrence.

**Funding acquisition:** Mary Donato, Beth A. Lawrence.

**Investigation:** Mary Donato, Olivia Johnson, Blaire Steven.

**Methodology:** Mary Donato, Olivia Johnson, Blaire Steven.

**Resources:** Beth A. Lawrence.

**Supervision:** Olivia Johnson, Beth A. Lawrence.

**Visualization:** Olivia Johnson, Blaire Steven.

**Writing – original draft:** Mary Donato, Blaire Steven.

**Writing – review & editing:** Mary Donato, Olivia Johnson, Blaire Steven, Beth A. Lawrence.

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
