## [Decision Letter · Decision Letter 0]

2 Mar 2020

PONE-D-20-02931

Nitrogen enrichment stimulates wetland plant responses whereas salt amendments alter microbial communities and biogeochemical responses

PLOS ONE

Dear Dr. Lawrence,

Thank you for submitting your manuscript to PLOS ONE. After careful consideration, we feel that it has merit but does not fully meet PLOS ONE’s publication criteria as it currently stands. Therefore, we invite you to submit a revised version of the manuscript that addresses the points raised during the review process.

In particular, criteria 3 and 4 (https://journals.plos.org/plosone/s/criteria-for-publication) have not been fully achieved. Experimental and statistical issues were raised by both reviewers, which need to be addressed by the authors as well as it is crucial to verify the values of tables/figures and text to avoid any misleading information. 

I also have few additional points:

- Have the authors considered the distinction between heterotrophs/organotrophs and heterotrophs/lithotrophs, when presenting the results in lines 416-419? These distinctions, in combination with data on salt tolerance could bring more light further into the discussion. These characteristics should then be included in table S1 and should also be used in the discussion of the results (lines 482-503).

- The authors in the material and methods section clearly indicated that bacterial communities were not collected from the rhizosphere, but from bulk soil from the root zone. I believe the discussion in lines 473-481 would be improved if the authors also related the results with known effect of soil properties/soil parent material in the bacterial communities.

Some minor points:

Line 384-388, besides, the genus of the identified OTUs, also indicate in brackets the phylum;

Figure 8, in particular in Figure 8a, please change (Average counts treatment (log10)) by (ratio of average counts treatment (log10), controls versus treatment);

Line 456 replace “root porocity” by “root porosity”.

We would appreciate receiving your revised manuscript by Apr 16 2020 11:59PM. To enhance the reproducibility of your results, we recommend that if applicable you deposit your laboratory protocols in protocols.io, where a protocol can be assigned its own identifier (DOI) such that it can be cited independently in the future. For instructions see: http://journals.plos.org/plosone/s/submission-guidelines#loc-laboratory-protocols

We look forward to receiving your revised manuscript.

Kind regards,

Ana R. Lopes, PhD

Academic Editor

PLOS ONE

Journal Requirements:

5. Our internal editors have looked over your manuscript and determined that it is within the scope of our Call for Papers on the Microbial Ecology of Changing Environments. Additional information can be found on our announcement page: https://collections.plos.org/s/microbial-ecology. If you would like your manuscript to be considered for this collection, please let us know in your cover letter and we will ensure that your paper is treated as if you were responding to this call. If you would prefer to remove your manuscript from collection consideration, please specify this in the cover letter.

6. In your Methods section, please provide additional information regarding the permits you obtained for the work. Please ensure you have included the full name of the authority that approved the field site access and, if no permits were required, a brief statement explaining why.

Reviewers' comments:

Reviewer's Responses to Questions

**Comments to the Author**

1. Is the manuscript technically sound, and do the data support the conclusions?

Reviewer #1: Yes

Reviewer #2: Yes

2. Has the statistical analysis been performed appropriately and rigorously? 

Reviewer #1: No

Reviewer #2: Yes

3. Have the authors made all data underlying the findings in their manuscript fully available?

Reviewer #1: Yes

Reviewer #2: Yes

4. Is the manuscript presented in an intelligible fashion and written in standard English?

Reviewer #1: Yes

Reviewer #2: Yes

5. Review Comments to the Author

Reviewer #1: Here Donat et al. analyzed the impacts of salt and nitrogen deposition on carbon cycling in wetland monocultures. They used 48, ~380 L tanks to make 4 replicates of 3 different species with 4 different water chemistry treatments. Overall the authors make a strong case for how salts impact both vegetation and microbial communities and thus the carbon cycling in wetlands. The study is original but some work needs could be done to contextualize the results within the broader literature to help expand the relevance of the mesocosm study the authors present (see papers below). This is particularly true when discussing the relevance of the OTU abundance on carbon cycling. Salt amendments lowered C mineralization, and altered OTU diversity, but the link between the two needs to be more carefully addressed within the discussion. Finally, the authors need to address the design components which could have impacted the conclusions.

It is unclear if the authors consider the impacts of draining the tanks on the reduction-oxidation potential of the micropores within the soil? Although the seasonal methane signal was not described in the text, the relatively large p-value suggests that methane may not have been consistent across all sampling periods. Could exposure of the soil matrix to the atmosphere be part of the story? The authors should address this.

The Gibbs free energy that determines which terminal electron acceptor is the dominant metabolic pathway is pH dependent (See Bethke et al. 2011). If the soils and treatments varied in acidity, particularly around neutral conditions, metal reduction rates could be more relevant than the SO4 content in the context of limiting methanogenesis. If the pH wasn’t measured, then the authors need convince the reader that the pH is similar across all treatments or they need to soften their language around ‘favorablitily’ of the kinetics that regulate microbial metabolisms.

The statistics reported throughout the paper need some improvement. In one case the authors accept a null hypothesis with a p-value of 0.077 (line 222), but then reject it in separate analyses with a p-value of 0.078 (Ln 267) and 0.069 (Ln 269). Liner mixed effect models were poorly described but used to justify aggregating methane fluxes despite relatively low p-values. This may be acceptable, or may be problematic depending on the specific hierarchal model design. The correlation analyses could also use some clarification, since it was unclear when Pearson correlation were used or when the Spearmen rank test was implemented. Clarity would help the reader understand the decisions around test selection, as it was initially assumed by this reviewer that the log transform was done to normalize the data. Why then use a Spearmen rank test in this instance, or if using Spearmen, why log transform the data? Furthermore, regression analysis may be more informative than Pearson in a number of analyses presented here, since it allows for multivariate analysis and post hoc analysis of residuals. For example, a properly constructed linear model could determine if the correlation between SO4 and methane is independent of the Spartina group, which can not be inferred from the Person correlation analysis presented in the text.

Recommended Papers:

Granberg, G., Sundh, I., Svensson, H., and Nilsson, M., Effects of temperature and nitrogen and sulfur deposition on methane emission from a boreal mire, Ecology, 82, 1982–1998, 2001.

Bethke, C. M., Sanford, R. A., Kirk, M. F., Jin, Q., and Flynn, T. M., The thermodynamic ladder in geomicrobiology, Am. J. Sci., 311, 183–210, 2011.

Herndon, E. M., Mann, B. F., Roy Chowdhury, T., Yang, Z., Wullschleger, S. D., Graham, D., Liang, L., and Gu, B., Pathways of anaerobic organic matter decomposition in tundra soils from Barrow, Alaska, J. Geophys. Res.-Biogeo., 120, 2345– 2359, 2015.

Christiansen, J. R., Levy-Booth, D. J., Prescott, C. E., and Grayston, S. J., Microbial and environmental controls of methane fluxes along a soil moisture gradient in a Pacific coastal temperate rainforest, Ecosystems, 19, 1255–1270, 2016.

Gao, C., Sander, M., Agethen, S., and Knorr, K.-H., Electron accepting capacity of dissolved and particulate organic matter control CO2 and CH4 formation in peat soils, Geochim. Cosmochim. Ac., 245, 266–277, 2019.

Clark, M. G., Humphreys, E. R., Carey, S. K., Low methane emissions from a boreal wetland constructed on oil sand mine tailings, Biogeosciences, 17, 667-682, 2020.

Specific comments:

Line 84: Maybe some reclamation lit here?

Line 53: Are they highly productive, or do they just have low rates of respiration? The 5th assessment report WG1 is not a good source for suggesting that wetlands are becoming more monotypic graminoid dominant.

Line 143: A schematic, of field picture to help visualize the experimental design would be helpful. Even if it’s in the supplementary data.

Line 161: What is the accuracy on the allometric equations?

Line 273: Imprecise wording. I think it means it is relative to the control and the salt treatments? The control isn’t really a “treatment”.

Lines 290/91: This sentence is confusing. Do you simply mean there was a species and treatment effect but no interaction term? Simple wording clarification should be fine.

Line 294/95: No mention the road salt treatment.

Line 304: There was an issue with the µ (/mu) symbol on the pdf.

Line 441: ends with a citation, but is written as if it was a result from this study.

Lines 429-430 is vague wording. Is it largely similar in response to water quality treatment or largely similar in the differences across species within each treatment?

Line 438-40: “environmentally relevant” and “we targeted” is redundant here. You already explained in the methods that you targeted environmentally relevant salinity profiles.

Line 440-442: Did you mean “Other studies have found…”?

Line 442-443: Did Walker S. (2019) find only an effect at >2 ppt? Your wording leaves their results ambiguous. Did they find no effect at <2 or didn’t include it in their study?

Line 447: “across all treatment” sounds like correlation analysis was performed within each treatment but was not reported.

Lines 450-452: The correlation to DOC is a separate idea. Put it in a new sentence for clarity unless it was only negatively correlated in the salt treatments.

Line 453: Was the relationship between microbial community abundance and heterotrophic respiration examined, why wasn’t it reported? I would assume some of these communities would have no impact on aerobic respiration.

Line 457: reference reads as if its on your own findings. Perhaps add “…as others [ref] have described”?

The paragraph starting on Line 455 is difficult to follow. A lot of ideas that jump around and don’t build on one another. Reorder the SO4-Spartina discussion so it builds to the conclusion, that methane is largest in these plots. The second sentence states that “Our data shows” but has two citations. You need to say what specifically in the other studies relate to what your data shows.

Lines 468/69: Why “this is surprising” is unclear, since expectations for production/emission by species was never discussed. Is the surprise simply referring to the productivity, the methane flux, or both?

Line 483: Is this in reference to Fig 6b, not 4b?

Line 487: Again, I think this is Fig 8C. Fig 6A has nothing to do with rarity of populations. This sort of repeated mistake makes it hard to follow the discussion and has me wonder what else has been overlooked.

Line 492: I am not a microbiologists. For those who share this shortcoming, I wonder if you could add the primary metabolic pathway (i.e. sulphate reducers, methanogens, etc.) of the microbes to table S1 or perhaps list it when you first discuss the Phyla of the microbes? It would help readers follow the logic in the discussions on species abundances and reinforce the connection to the carbon cycle.

Line 503: Did you look into composition/abundance and mineralization rates? I didn’t see it in your manuscript but could be another link between microbes and the carbon cycle.

Discussion: What are the implications of reduced fPAR on the carbon cycle with respect to your treatments? It was not included in the discussion, so is it relevant to the study?

Readability of Figure 8 is very low. Impossible to see axis labels. Increase resolution.

Reviewer #2: Donato and collaborators are describing the carbon fluxes and microbial changes in wetland mesocosms exposed to nitrogen, road salt, or sea salt contaminations. They showed that carbon fluxes were mainly impaired by Nitrogen treatment through plant biomass changes, the salt pollution disturbed C mineralization, decrease in microbial diversity and shifts in the microbial community, and led to lower CH4 emissions.

The authors have made a great effort in compiling a complex dataset in a very comprehensive study. The study is detailed, and the results well explained. The conclusions are interesting and will be of interest to the PLOS ONE readers.

My main comments would be that the authors do not discuss the eutrophication of their system, neither the (likely high) reducing environment that the N and salt treatment likely generated. Though an additional discussion of that matter would be interesting, it should not prevent the publication of this paper that is already convincing as it is. See my other comments below:

1. Was anything added to the control treatment (eg 1L of DIW?)

2. Line 162: Please provide a reference or in SI the correlations used to calculate the plant biomass based on their height.

3. It would be of interest to specify that the sampling campaigns were done 1months, 2 months, and 3 months after the last dosing treatment, during the summer (right?)

4. Line 286: significances between treatments or between plant species? Line 280 = treatments do not impact root porosity)

5. All plot, I would call the panels “treatments” rather than “Water quality”, which is a trait itself. It is a bit confusing as it.

6. Line 303 and fig 3a: the number provided in the text vs the values plotted doesn’t seem to match.

7. It has been shown in similar systems that redox variations can vary daily and along the year1, and be impacted by anthropogenic activities. These effects were likely mainly driven by the macrophyte photosynthesis, respiration, and life cycles. I am pointing this because based on my (rough) calculations, the authors have added approximately 5g of SO42- in their system (based on the characterization provided in ref 41 in the manuscript), which should represent a concentration in the 20L mesocosms of 250 mg/L. Despite the very high addition of SO42- added to their system the authors measured “only” 3.2 mg SO42- per L of water in their mesocosms. This likely indicates a reducing environment (which is not surprising since the sampling was done during the summer, as described in ref1).

My question is: could it be that some of the observed effects in the sea salt treatment (decrease of DOC, C mineralization) are due to a more reducing environment than the others? Did the authors measure dissolve CO2 and O2 concentrations in the mesocosms water?

8 Line 485: it could be osmotic stress and/or a more reducing environment.

9 Line 463: Spartina is also very responsive to the nitrogen treatment. Could it be an indicator of the eutrophication of the system, thus more reduction of the water, explaining the larger decrease in SO42- concentration? I am surprised that the authors do not discuss eutrophication here when implementing their systems with N at a very high rate (15 g N /year)

10 Line 491:Figure 5 also indicates that the 2salts treatments induce a shift toward the CCA1 factor, and the 2 populations are grouped. This could indicate that, at that period of the year (already under pretty high reducing conditions), the microorganisms are more sensitive to high salinity than to the redox conditions.

11 I could be of interrest for the reader to add a picture of the mesocosms in supporting information

Cited reference

1 Andersen, et al. "Extreme diel dissolved oxygen and carbon cycles in shallow vegetated lakes." Proceedings of the Royal Society B: Biological Sciences 284.1862 (2017): 20171427.

2 Simonin, et al. "Engineered nanoparticles interact with nutrients to intensify eutrophication in a wetland ecosystem experiment." Ecological Applications 28.6 (2018): 1435-1449.

6. PLOS authors have the option to publish the peer review history of their article (what does this mean?). If published, this will include your full peer review and any attached files.

Reviewer #1: No

Reviewer #2: No

---

## [Author Response · Author response to Decision Letter 0]

23 Apr 2020

Please see our uploaded "Response to Reviews" where we detail how we addressed all editor and reviewer suggestions.

---

## [Decision Letter · Decision Letter 1]

18 May 2020

PONE-D-20-02931R1

Nitrogen enrichment stimulates wetland plant responses whereas salt amendments alter sediment microbial communities and biogeochemical responses

PLOS ONE

Dear Dr. Lawrence,

Thank you for submitting your manuscript to PLOS ONE. After careful consideration, we feel that it has merit but does not fully meet PLOS ONE’s publication criteria as it currently stands. Therefore, we invite you to submit a revised version of the manuscript that addresses the points raised during the review process.

The authors addressed most of the previous reviewer’s concerns, however some minor points need to be addressed. Please modify the manuscript according to reviewers’ suggestions. Moreover, use the same format for units all over the manuscript (including figures) (mg/L or μmol m^-2^ h^-1^).

We would appreciate receiving your revised manuscript by Jul 02 2020 11:59PM. To enhance the reproducibility of your results, we recommend that if applicable you deposit your laboratory protocols in protocols.io, where a protocol can be assigned its own identifier (DOI) such that it can be cited independently in the future. For instructions see: http://journals.plos.org/plosone/s/submission-guidelines#loc-laboratory-protocols

We look forward to receiving your revised manuscript.

Kind regards,

Ana R. Lopes, PhD

Academic Editor

PLOS ONE

Reviewers' comments:

Reviewer's Responses to Questions

**Comments to the Author**

1. If the authors have adequately addressed your comments raised in a previous round of review and you feel that this manuscript is now acceptable for publication, you may indicate that here to bypass the “Comments to the Author” section, enter your conflict of interest statement in the “Confidential to Editor” section, and submit your "Accept" recommendation.

Reviewer #1: All comments have been addressed

Reviewer #2: All comments have been addressed

2. Is the manuscript technically sound, and do the data support the conclusions?

Reviewer #1: Yes

Reviewer #2: Yes

3. Has the statistical analysis been performed appropriately and rigorously? 

Reviewer #1: Yes

Reviewer #2: Yes

4. Have the authors made all data underlying the findings in their manuscript fully available?

Reviewer #1: Yes

Reviewer #2: Yes

5. Is the manuscript presented in an intelligible fashion and written in standard English?

Reviewer #1: Yes

Reviewer #2: Yes

6. Review Comments to the Author

Reviewer #1: This version is much stronger submission. Donato et al. clearly structure and organize their explanation of the impacts of nitrogen and salt treatments on wetland mesocosm’s. The results, attempting to bridge fields in biogeochemistry, botany, ecology, and microbiology are relevant and topical. The discussion could be extended to wider literature, but I do not believe that should prevent the publication of this manuscript. I also appreciate that the limitations are now clearly addressed in their own section within the manuscript. I believe this manuscript would be of interest to the readers of PLOS ONE. I have a few very minor comments below.

General:

Some figures have the first sentence of the caption bolded, some do not (Figures 3 & 5).

Microbial community response results are clearly tied to the rest of the manuscript in this draft, but as currently written some of the “results” would be better suited for the discussion section.

Specifics:

Line 222: Clearly state what the random effects are, I have assumed individual gas flux plot location (i.e. repeated measures).

Line 224: lme stands for Liner Mixed-Effect Model, although an LME is an acceptable way to construct a repeated measures ANOVA, I suggest that since you started with the mixed effect language above (Line 222) you should stick to one terminology for clarity.

Line 227 states that “only the September gas sampling data is used” when investigating relationship with other “variables” and Line 229/30 states “tested for correlations across the entire data set”. This language is imprecise. I believe all 3 gas sampling campaigns were used for carbon flux analysis between species and water treatments, and the limited September data set is used for correlation analysis with pore water chemistry described under “Correlations with carbon fluxes”. In this section you should just clearly state what data set is used when to stop the reader from making assumptions about your statistical design.

Lines 270-274: There is a lot of information in this sentence, can it be broken up into two separate sentences for improved readability.

Line 577/78: The conclusion is not the place for this new reference to nanoparticles.

Reviewer #2: I think this reviewed manuscript is a good improvement of the initial submission. Clarifying the choices about the statistical tests, along with a more clear methodology makes the paper stronger. I agree with the authors that discussing microbial function based on their diversity and abundance data may be overstepped. I like the new section “experimental design constrains” that the authors now provide, and that is central in the experimentation on analogue models

My last minor comments are:

- authors stated that their sampling was done away from the root influence. Their results may suggest otherwise, as for instance the Shannon’s diversity index significantly varied with the vegetation.

- It is also unclear to me why the authors have made 3 samplings for CO2 uptake rates and CH4 emissions, when they did only one harvesting for the others.

- The authors haven’t addressed the questions about the (likely occurring, but not measured) pH and oxidative state changes that could be influenced by time, the different treatments and the experimental setup (draining before winter). It needs to be acknowledge somewhere that they could be factors explaining the biogeochemical change observations

7. PLOS authors have the option to publish the peer review history of their article (what does this mean?). If published, this will include your full peer review and any attached files.

Reviewer #1: No

Reviewer #2: No

---

## [Author Response · Author response to Decision Letter 1]

4 Jun 2020

We address all reviewer and editor comments in our attached "Response to Reviews" document.

---

## [Decision Letter · Decision Letter 2]

11 Jun 2020

Nitrogen enrichment stimulates wetland plant responses whereas salt amendments alter sediment microbial communities and biogeochemical responses

PONE-D-20-02931R2

Dear Dr. Lawrence,

We’re pleased to inform you that your manuscript has been judged scientifically suitable for publication and will be formally accepted for publication once it meets all outstanding technical requirements.

Kind regards,

Ana R. Lopes, PhD

Academic Editor

PLOS ONE

Additional Editor Comments (optional):

Reviewers' comments:

Reviewer's Responses to Questions

**Comments to the Author**

1. If the authors have adequately addressed your comments raised in a previous round of review and you feel that this manuscript is now acceptable for publication, you may indicate that here to bypass the “Comments to the Author” section, enter your conflict of interest statement in the “Confidential to Editor” section, and submit your "Accept" recommendation.

Reviewer #2: All comments have been addressed

2. Is the manuscript technically sound, and do the data support the conclusions?

Reviewer #2: Yes

3. Has the statistical analysis been performed appropriately and rigorously? 

Reviewer #2: Yes

4. Have the authors made all data underlying the findings in their manuscript fully available?

Reviewer #2: Yes

5. Is the manuscript presented in an intelligible fashion and written in standard English?

Reviewer #2: Yes

6. Review Comments to the Author

Reviewer #2: I think the authors have addressed all the reviewer's comments and suggestions. I recommend that the manuscript is accepted for publication as it is.

7. PLOS authors have the option to publish the peer review history of their article (what does this mean?). If published, this will include your full peer review and any attached files.

Reviewer #2: No

---

## [Editor Report · Acceptance letter]

1 Jul 2020

PONE-D-20-02931R2 

Nitrogen enrichment stimulates wetland plant responses whereas salt amendments alter sediment microbial communities and biogeochemical responses 

Dear Dr. Lawrence:

I'm pleased to inform you that your manuscript has been deemed suitable for publication in PLOS ONE. Congratulations! Your manuscript is now with our production department. 

Kind regards, 

on behalf of

Dr. Ana R. Lopes 

Academic Editor

PLOS ONE